# Nontrivial nanostructure, stress relaxation mechanisms, and crystallography for pressure-induced Si-I → Si-II phase transformation

Hao Chen[1], Valery I. Levitas [2,3 ✉], Dmitry Popov[4 ✉] & Nenad Velisavljevic[4,5]

Crystallographic theory based on energy minimization suggests austenite-twinned martensite interfaces with specific orientation, which are confirmed experimentally for various materials. Pressure-induced phase transformation (PT) from semiconducting Si-I to metallic Si-II, due to very large and anisotropic transformation strain, may challenge this theory. Here, unexpected nanostructure evolution during Si-I → Si-II PT is revealed by combining molecular dynamics (MD), crystallographic theory, generalized for strained crystals, and in situ real-time Laue X-ray diffraction (XRD). Twinned Si-II, consisting of two martensitic variants, and unexpected nanobands, consisting of alternating strongly deformed and rotated residual Si-I and third variant of Si-II, form {111} interface with Si-I and produce almost self-accommodated nanostructure despite the large transformation volumetric strain of −0.237. The interfacial bands arrest the {111} interfaces, leading to repeating nucleation-growth-arrest process and to growth by propagating {110} interface, which (as well as {111} interface) do not appear in traditional crystallographic theory.

[1] Key Laboratory of Pressure Systems and Safety, Ministry of Education, School of Mechanical and Power Engineering, East China University of Science and Technology, Shanghai 200237, People's Republic of China. [2] Iowa State University, Departments of Aerospace Engineering and Mechanical Engineering, Ames, IA 50011, USA. [3] Ames Laboratory, Division of Materials Science and Engineering, Ames, IA, USA. [4] HPCAT, X-ray Science Division, Argonne National Laboratory, Lemont, IL, USA. [5] Physics Division, Lawrence Livermore National Laboratory, Livermore, CA 94550, USA. ✉email: vlevitas@iastate.edu; dpopov@anl.gov

One of the challenging goals in studying high-pressure PTs in materials is finding real time microstructure evolution. This includes crystallographic features of PT, orientation of interfaces, morphology of phases, and stress relaxation mechanisms. Crystallographic theory based on energy minimization[1,2] suggests austenite-twinned martensite interfaces with specific orientation, which are confirmed experimentally for various materials. For phases that do not exist in the stress-free state, including Si-II, in situ measurements are vital and essential in studying high-pressure PTs. High pressure Laue diffraction is a powerful tool to investigate microstructure evolution across PT in situ[3–5], however, a broader application of this powerful experimental capability also requires a strong collaboration with modeling and theory work. Martensitic PT from Si-I to Si-II occurring at 12–14 GPa is studied broadly[3,5–10]. Si-I/Si-II interface is observed but not uniquely indexed with *Laue* diffraction, and twining in Si-II is mentioned hypothetically by Popov et al.[5]. However, MD simulations[11–13] do not show twinned Si-II. MD simulations reveal atomic features of PT at nm-ps resolution. However, due to limitations of small process duration (ns), sample size (μm), and high strain rate (ps$^{-1}$), MD stress relaxation mechanisms and nanostructure may deviate from reality. XRD, due to spatial resolution of few micrometers and time resolution of few minutes may miss some finer features and transitional processes. Also, it does not allow unambiguous indexing of interfaces without some assumptions. Here, we show major agreement between these two approaches for the revealed counterintuitive nanostructure, which is very nontrivial and strongly supports both.

Crystallographic theory[1,2] for cubic-tetragonal PT suggests that there are three tetragonal martensitic variants of Si-II with transformational deformation gradients (i.e., for neglected elastic strains) in cubic coordinates $\mathbf{F}_t^1 = \{a; a; b\}$, $\mathbf{F}_t^2 = \{a; b; a\}$, $\mathbf{F}_t^3 = \{b; a; a\}$, where $a = 1.175$ and $b = 0.553$[11], i.e., transformation strains ($a - 1$ and $b - 1$) are large, including large transformation volumetric strain $e_v = \det\mathbf{F}_t - 1 = -0.237$. Each pair of Si-II variants are in twin relation with each other with a {110} twinning plane of Si-I lattice and very large twinning shear $\gamma = 1.655$[1]. Si-I and the mixture of two twin-related Si-II martensitic variants are compatible (i.e., satisfies the Hadamard compatibility condition that follows from the displacement continuity across a coherent interface) for an interface with unit normal $\mathbf{m} = [0.631; 0.754; 0.183]$. Also, complete self-accommodating mixture of martensitic variants (i.e., mixture that occupies region of the same size and shape as the parent phase) within austenitic matrix (e.g., in diamond-shape region),

which does not generate any long-range stresses, requires zero volumetric strain[1], i.e., it is impossible for Si-II.

In this work, we hypothesize that large strains may lead to microstructure and stress relaxation mechanisms, that do not follow traditional crystallographic theories. Each experimental and simulation method has its own pros and cons and cannot give a complete picture. We combine crystallographic theory (which we expand to the deformed crystals and complex nanostructures), MD, and in situ synchrotron radiation diffraction, and reveal unexpected nanostructure evolution during Si-I to Si-II PT, which contradicts the classical crystallographic theory. All three approaches revealed twinned Si-II with the same twinning planes {110}. Twinned Si-II, consisting of two martensitic variants, and unexpected nanobands, consisting of alternating strongly deformed and rotated residual Si-I and the third variant of Si-II, form {111} interface with Si-I and produce almost self-accommodated nanostructure despite the large transformation volumetric strain. The interfacial bands arrest the {111} interfaces, leading to repeating nucleation-growth-arrest process and to growth by propagating {110} interface, which (as well as {111} interface) do not appear in traditional crystallographic theory. Thus, in contrast to classical theory requiring stress-free interfaces, microstructure is governed by reduction of the long-range stresses in the entire volume, tolerating high short-range stresses at complex interfaces, which are necessary to keep residual Si-I and reduce resultant volume jump.

## Results

Computational scheme for MD simulations is presented in Fig. 1 and supplementary material. Si-I sample includes two dislocations, which cause stress concentration (Fig. 1), and is loaded by hydrostatic pressure. In the theory[14], single-variant martensite nucleates first and, after reaching some critical size, is then observed to twin. Here, twinned Si-II appears from the beginning (Fig. 1 and Supplementary Movie 6) near each dislocation. In XRD experiment twinning in Si-II is observed from small angles between <211> and <220> directions (Fig. 2 and Supplementary Movie 5), consistent with angle up to 8.3° obtained here with the crystallographic theory under strains. In MD simulations with a dislocation dipole (practically two independent single dislocations), pressure for initiation of Si-I → Si-II PT is lower than for dislocation-free crystal by a factor of 1.45. In reality, the defect-free Si-I should transform to Si-II at 18.3 GPa, when the first phonon instability in the first principle calculations is observed[9]. Then the observed PT pressure in the current experiment, ~13 GPa (see Supplementary Material), which matches the

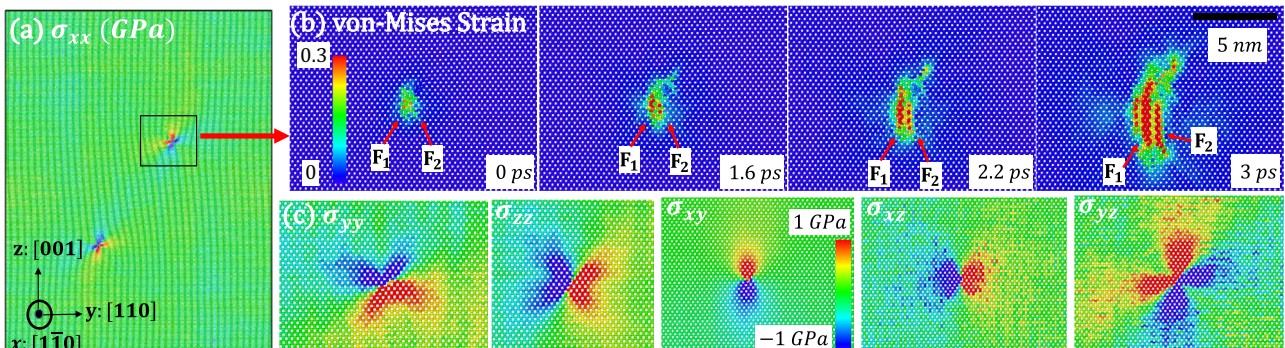

**Fig. 1 Field of internal stresses due to dislocations and nucleation of twinned Si-II at two dislocations. a** Computational model of a single Si crystal with shuffle 60° dislocation dipole inserted by employing dislocation displacement[27] at constant hydrostatic pressure, including internal stress distribution due to dislocations. Stress fields of dislocations practically do not overlap. **b** Evolution of von-Mises strain distribution. **c** Distribution of all internal stresses due to dislocation from MD simulation. The twined Si-II nucleates from single dislocation from the beginning and grows along the [110] (i.e., *y*) direction. The (110) twinning plane is consistent with the prediction from crystallographic theory[1] and the current experiment.

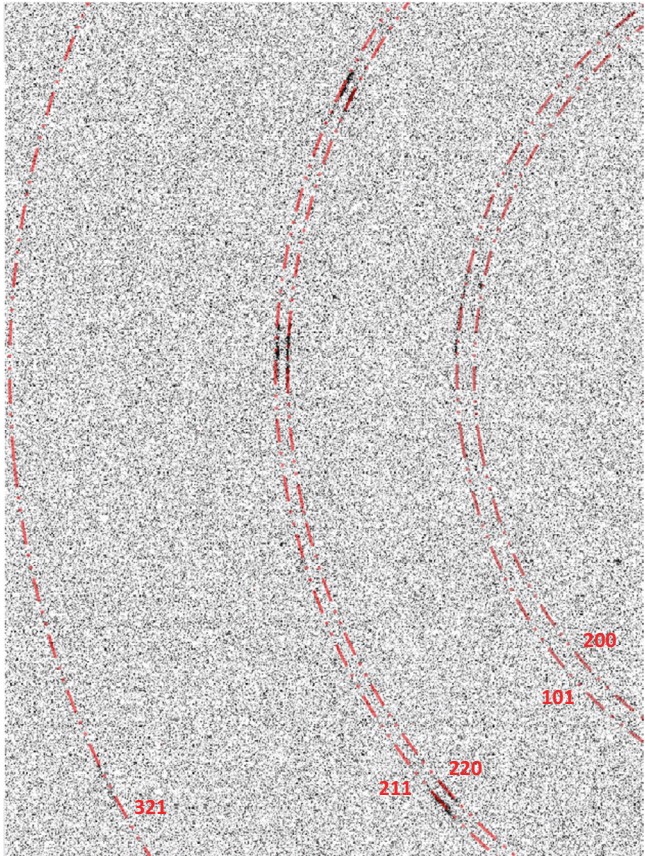

**Fig. 2 Oscillation diffraction pattern from Si-II obtained with monochromatic beam in 1° angular range.** Red circles denote predicted diffraction lines of Si-II.

typically reported pressures[3,5–10], is lower than 18.3 GPa by a factor of 1.41. Thus, a typical nucleation event in the experiment should occur at a single dislocation, consistent with low dislocation density in our Si-I sample. This contrasts with the statement in[9] that due to the large difference between calculated PT pressure 18.3 and 13 GPa in the experiment, phonon instability cannot be responsible for the initiation of PT; clearly, the effect of dislocations was missed. Another qualitative confirmation of the strong effect of defects on nucleation pressure in our experiments comes from the fact that there are irregular parts of a sample where nucleation did not occur at all because they are probably dislocation-free (see Supplementary Information).

After nucleation, nontrivial microstructure evolution is observed, as shown in Fig. 3 and Supplementary Movie 6. Both in MD and experiment (see below) Si-I and complex Si-II microstructure form a rational interface {111}, which smallest deviation from predicted by crystallographic theory normal $\mathbf{m}$ is 25.16°. In elasticity-based theory of martensite[15–17], a thin layer of alternating tips of twin-related $\mathbf{F}_t^1$ and $\mathbf{F}_t^2$ variants produce significant elastic energy. Due to very large twinning shear of 1.655 for Si-II as well as significant deviation of {111} interface from $\mathbf{m}$, the elastic energy relaxes by producing an unexpected interfacial nanoband II, consisting of alternating $\mathbf{F}_t^3$ variant of Si-II and strongly deformed and rotated residual Si-I (Fig. 3). Such an interfacial band was not observed for any PT in any material and represents nontrivial structural mechanism of internal stress relaxation. These interfacial bands II strongly reduce mobility of the twinned Si-II–Si-I interfaces {111}, which practically do not propagate. Microstructure evolves in two main ways (Fig. 3 and Supplementary Movie 6). (a) Lengthening of twinned Si-II and

interfacial bands along the bands in [11$\bar{2}$] direction. During this process, the bands form an interface between Si-I and Si-II close to (110) twin interface, which does not appear in crystallographic theory but is observed in our Laue diffraction experiment (see below). (b) New twinned Si-II nucleates and grows from an interfacial band II, producing next band I and then II. At the same time, previously formed bands I and II keep growing causing movement of (110) interface between Si-I and Si-II.

Using real-time Laue diffraction, we explicitly observed propagation of Si-I/Si-II interfaces, as projected onto a {110} plane (Fig. 4a, b and supplementary movies 1, 2, and Supplementary Information). The observed interfaces are parallel to [101] and [010] directions and, at the same time, the interfaces are tilted by large angles with respect to the plane of projection (10$\bar{1}$). It is important to stress that normal to the interface parallel to [101] is oriented far away from the normal $\mathbf{m}$ = [0.631;0.754; 0.183] predicted by crystallographic theory. The area of the newly formed Si-II overlaps with the area of the rest of Si-I indicating that at least one of these interfaces is tilted by angle of about 45° with respect to the plane of projection. Thus, possible orientations of Si-I/Si-II interfaces include {111} and {110} interfaces predicted by the MD simulations but do not exclude other directions. The Si-I/Si-II interface in projection on a {100} plane was observed in in situ Laue diffraction experiment[5], but it was not uniquely indexed and, apart from the current results, it was not clear whether Si-II areas grow. Thus, Si-II areas, as projected onto a {100} plane, are elongated parallel to a <110> direction indicating limited range of crystallographic planes, tilted by large angles with respect to the plane of projection and parallel to this <110> direction. Therefore, possible orientations of Si-I/Si-II interfaces also include {111} and {110} interfaces predicted by the MD simulations but also still do not exclude other directions. Results[5] did not reliably exclude interface with normal $\mathbf{m}$ = [0.631; 0.754; 0.183] predicted by crystallographic theory, in contrast to the current results. Si-II produces much broader and "streaky" reflections comparing to the parental phase. This is the indication of substantial misorientation of various nanodomains in Si-II, to some extent like those observed in Fig. 3. Since in[5] and here the same Si-I/Si-II interfaces are observed as projected onto different crystallographic planes, combining results can yield 3D orientations of the interfaces based solely on the experimental results under the following assumption. Since only interfaces parallel to rational axes with indices 0 or 1 have been observed, it is reasonable to suppose that the interfaces are parallel to rational crystallographic planes with the same kind of indices. This yields interfaces parallel to {110} and {111} predicted by the MD. As predicted, twinned Si-II and interfacial bands grow along the bands in [12$\bar{1}$] direction causing shift of the Si-I/Si-II interfaces parallel to (101) and (1$\bar{1}$1) (Fig. 4c). While interfacial nanobands so far have not been confirmed in experiment due to very small size and severe distortions from cubic Si-I and tetragonal Si-II lattices, experimental confirmation of unexpected {111} and {110} interfaces, which are directly related to them, makes this unexpected evolving nanostructure plausible.

## Discussion

To better understand the reasons for the unusual {111} interface, new interfacial bands, presence of residual Si-I deeply in the region of stability of Si-II, and two-band structures with specific spacing, we expanded traditional stress-free crystallographic theory for finite elastic strains (stresses) and evaluated degree of violation of the averaged Hadamard compatibility condition for each of the interface. As a measure of violation, we evaluate the principal components of 2D incompatibility strain tensor $\mathbf{Inc}$ (defined in Supplementary Material) that deform {111} interfaces between different phases or bands. For compatible interface

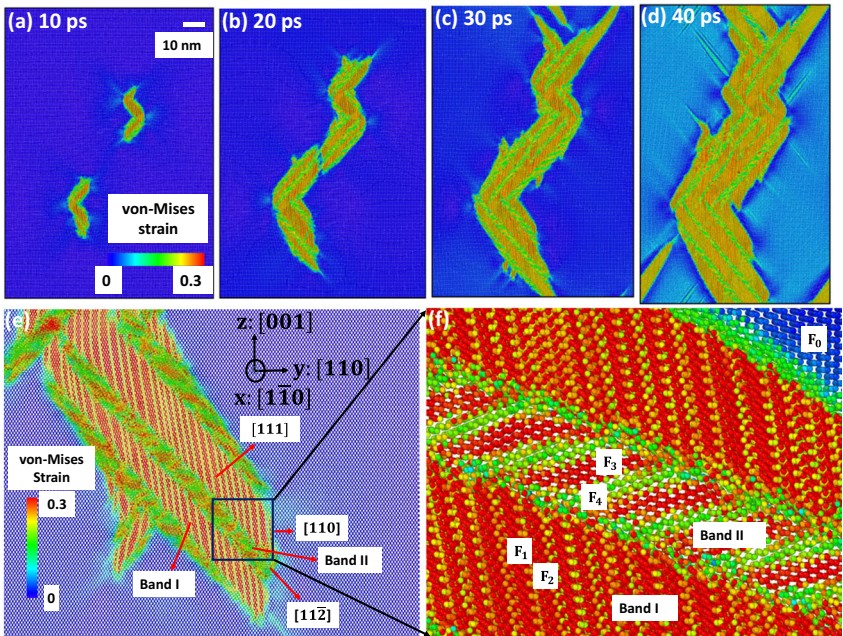

**Fig. 3 Snapshot of Si-II microstructure and its rotated and enlarged views. a–d** The evolution of the microstructure growing from two dislocation-induced nuclei. **e** and **f** The zoomed microstructures composed of the band I, consisting of alternating Si-II variants with deformation gradients $\mathbf{F}_1$ and $\mathbf{F}_2$ (which include elastic strains), separated by (110) twinning plane, and interfacial band II, consisting of alternating Si-II variant $\mathbf{F}_3$ and strongly deformed Si-I with deformation gradient $\mathbf{F}_4$. The interfaces between the two bands and between Si-I and band II are (111) planes. Another interface between twinned Si-II and Si-I is (110); both are not present in the crystallographic theory[1]. Growth occurs by propagation of (110) interface along the bands in [11$\bar{2}$] direction and by nucleation and growth of new twinned Si-II from an interfacial band II, producing next band I and then band II, as observed in the experimental measurement.

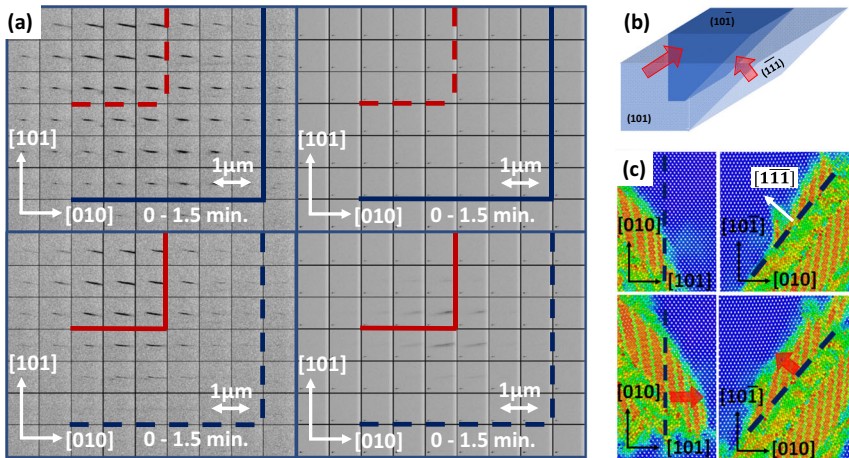

**Fig. 4 Shifts of Si-I/Si-II interfaces as observed by in situ real-time Laue diffraction and MD simulation. a** Maps of 202 reflection from a Si-I crystal (left column) and maps of a diffuse reflection from Si-II (right column). More maps are available in the Supplementary Movie 1. **b** 3D schematic of the shifts of the Si-I/Si-II interfaces. **c** Simulated shifts of the Si-I/Si-II interfaces. In all the images the interfaces before the shifts are shown by dark blue and the interfaces after the shifts are shown by red lines; solid lines denote currently existing interfaces and dotted dashed lines denote interfaces existing in different states; directions of the shifts of the interfaces are shown by red arrows.

**Inc** = (0; 0). Thus, for the band I-Si-I interface, **Inc** = (0.069; 0.087), i.e., corresponding strains are finite (i.e., on the order of 0.1), which along with local stresses due to alternating twin tips cause nucleation of the band II. For the band II-Si-I interface, **Inc** = (0.117; 0.258), which causes much higher local stresses. For the band I-band II interface, **Inc** = (0.326; 0.201), which leads to huge local stresses. Thus, incompatibility analysis only increased perplexity: why does the system choose such sophisticated highly energetic nanostructure? However, these interfacial stresses are short-range and partially relaxed by local atomic rearrangements and loss of coherence

(Fig. 3b). Most importantly, (a) averaged over the bands I and II deformation gradient ($\mathbf{F}_{av} = \lambda_i \mathbf{F}_i$ with volume fractions $\lambda_1 = \lambda_2 = 0.3742, \lambda_3 = 0.1740, \lambda_4 = 0.1126$) produces with Si-I very small incompatibility **Inc** = (0.004; 0.021); (b) normal to the {111} interface strain $\mathbf{F}_{av} - \mathbf{F}_0$ is also small (−0.032); (c) difference in volumetric deformation gradients $\det\mathbf{F}_{av} = 0.718$ and $\det\mathbf{F}_0 = 0.763$ is −0.045, i.e., small (in comparison with $\epsilon_v = 0.237$) as well. All these facts produce almost self-accommodated bands I + II nanostructure with small averaged strain within Si-I matrix, and consequently, small long-range internal stresses. The known self-accommodated diamond

microstructure in unstressed shape memory alloys requires zero transformation volume strain and is produced by twin-within-twin structures of the martensite only. Since conditions here are very different (finite elastic deformation of both phases, large transformation volumetric strain $\epsilon_v = 0.237$, very large twinning shear of 1.655), this leads to completely different almost self-accommodated bands I + II nanostructure, which tolerates significant local interfacial incompatibilities, and involves strongly distorted metastable Si-I with $\mathbf{F}_4$ to reduce volumetric difference with hydrostatically deformed Si-I. This determines specific spacing (relative volume fraction) of bands I and II and their unusual {111} interfaces. Thus, huge internal stresses, in particular caused by third variant, are necessary to retain Si-I deeply in the region of stability of Si-II. This mechanism is consistent with the later stage of the PT (Fig. S1 and Supplementary Movie 6): with reducing volume fraction of remaining Si-I matrix and the elastic constraint due to the matrix, volume fractions of band II and of the variant $\mathbf{F}_3$ ($\lambda_3$) reduce, and after Si-I matrix disappeared, band II disappears as well leaving twinned Si-II and relaxing elastic stresses. Significant relaxation of lattice distortions in Si-II after completing PT was observed with XRD as well, indicated by "streaky" Laue and arch-like monochromatic beam reflections (Fig. 2). Thus, despite very different time and space scales in MD simulations and experiments, they show surprising and very nontrivial agreement in microstructure and crystallography.

Strong local stress concentrator at the intermediate band II in Si may be a cause for complex polymorphism of Si, leading under different types of loadings and defects to different phases (Si-III, IV, V, VIII, IX, XI, and XII, as well as amorphous Si). Understanding and controlling these stresses and nanostructure may lead to controlling selection of desired known and new (hidden) phases. Obtained combined in situ experimental, MD, and theoretical approaches open opportunities to study other high-pressure PTs in Si and Ge and other materials with large transformations strains, e.g., C, BN, and BCN systems (e.g., −0.39 for PT graphite to diamond or hexagonal to cubic or wurtzitic BN). Obtained results also challenge modern phase field approaches[18–20] so that they can describe the revealed nanostructures.

## Methods

**Simulation method**. In this work, classical MD simulations were performed using the LAMMPS package[21]. Simulations have been carried out at two constant temperatures, 1 and 300 K, using the Nosé–Hoover thermostat, and for multiple geometries; the obtained nanostructure and even pressure for initiation of PT were the same. The employed interatomic force field for the interactions between Si atoms was from the Tersoff interatomic potential[22]. This potential has been demonstrated to be successful in describing the crystal structure transition from the diamond-cubic to β-Sn in single crystal silicon (Si-I to Si-II) under a uniaxial stress of ~ 12 GPa[10], which is close to the experimental value[5–7]. The advantage of the Tersoff interatomic potential for the description of Si-I to Si-II phase transformation in comparison with four other potentials is demonstrated in[12].

We are interested in introducing a single dislocation as a nucleation site for the initiation of the Si-I–Si-II PT. However, to minimize its effect on the periodic boundary conditions at the boundaries, dislocation dipole must be placed with equal in magnitude but opposite Burgers vectors[23,24]. To reduce the effect of one dislocation on the stress concentrator from another one, distance between dislocations should be maximized. Figure 1a shows the computer model set-up of a single crystalline silicon with static shuffle 60° dislocation dipole with Burgers vector $b = 1/2[1\bar{1}0]$ inserted by imposing the displacement field of dislocations[13]. The dislocation dipole was put on the diagonal of the $yz$ plane to maximize the distance of the two dislocations. The dislocation internal stress field is shown in Fig. 1a, c and is consistent with theoretical predictions[13]. Periodic boundary conditions are applied along all three cubic directions to exclude free surface effect. One of the simulation cells has dimensions of $L_y \approx 99.7$ nm and $L_z \approx 130$ nm and the distance between dislocations of about 82.01 nm. The length $L_x$ has been varied at 1 K from 4 to 30 nm, with a sample containing around 2.5 million to 15 million atoms, and the results are found to be independent of this length due to the periodic boundary conditions applied. Another simulation cell has dimensions of $L_y \approx 199.4$ nm, $L_z \approx 260$ nm, and $L_x \approx 4$ nm, with the distance between dislocations of 164.02 nm containing around 10 million atoms and the

microstructure and the obtained at 300 K nanostructure and pressure for initiation of the PT we the same as for smaller sample and 1 K. Thus, the effect of one dislocation on the nucleation processes at another dislocation is negligible, and our results can be interpreted as nucleation and growth at a single dislocation. The pressure was applied to the system using the Berendsen algorithm[25], in which the instantaneous stress of the system was calculated using the virial formula.

**Experimental method**. High pressure *Laue* diffraction experiments have been conducted using experimental setup available at 16 BMB beamline of Advanced Photon Source[3]. Incident polychromatic beam, with the highest X-ray energy limit about 90 keV, reached the sample through one of the diamonds while the diffracted beams reached Perkin Elmer area detector, positioned at about 600 mm from the sample and tilted vertically by 30°, through the other diamond. X-ray incident beam was focused using *KB*-mirrors down to $3 \times 3\,\mu m^2$ at the half width of beam profiles. The sample was cut manually from the same Si wafer (University Wafer[26]) as for the previous research[5] and had dimensions about $15 \times 40\,\mu m^2$. The sample was put into a diamond anvil cell (DAC), having total opening of 60°, such that the nearly flat surface parallel to a {110} plane was also parallel to plane of one of the diamond anvils. The DAC was tilted vertically by 25° providing reasonable number of reflections for indexation. Series of two dimensional (2D) translational scans were collected on the sample across the transition with vertical translations along [101] direction parallel to the longest sample dimension. The 2D scans were collected on an area within the sample heaving size $10 \times 10\,\mu m^2$ with 1 μm step in both directions, in 2.5 min each. Precise sample orientation with respect to the translational axes was determined at pressure right below the PT by indexation of diffraction patterns: [101] direction of the sample deviated by about 12° from the vertical axis of translation and [010] direction of the sample deviated by about 3° from the horizontal axis of translation. Therefore, all the details of PT were projected onto a {110} plane with vertical axis of translation nearly parallel to a <110> direction and horizontal axis of translation nearly parallel to a <100> direction. Other details of experiment and data analysis are available in supplementary materials.

## Data availability

The data that support the findings of this study are available from the corresponding authors upon request.

## Code availability

Software polyLaue to analyze Laue diffraction data is available from its author upon reasonable request. Molecular dynamics simulation software LAMMPS is available from its website.

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

## Acknowledgements

H.C. acknowledges support by NSFC of China (52005186) and Shanghai Sailing Program (20YF1409400). VIL work was sponsored by NSF (CMMI-1943710, MMN-1904830, and XSEDE TG-MSS170015), ONR (N00014-16-1-2079), and ISU (Vance Coffman Faculty Chair Professorship). DP and NV acknowledge High Pressure Collaborative Access Team (HPCAT) (Sector 16), Advanced Photon Source (APS), Argonne National Laboratory. HPCAT operations are supported by DOE-NNSA's Office of Experimental Sciences. The Advanced Photon Source is a U.S. Department of Energy (DOE) Office of Science User Facility operated for the DOE Office of Science by Argonne National Laboratory under Contract No. DE-AC02-06CH11357. NV work is performed under the auspices of the U.S. Department of Energy by Lawrence Livermore National Laboratory under Contract DE-AC52-07NA27344.

## Author contributions

H.C. performed computational and theoretical work. V.L. supervised computational and theoretical work and performed connection between theory and experiments. D.P. collected and analyzed X-ray diffraction data. V.L., D.P., H.C., and N.V. wrote the manuscript.

## Competing interests

The authors declare no competing interests.
