## [Peer Review File · Nature Communications]

Reviewers' Comments:

Reviewer #1:

Remarks to the Author:

Thank you for the opportunity to review the manuscript by Chen and colleagues. The paper describes a detailed investigation of the Si-I to Si-II pressure-induced phase transformation through a combination of molecular dynamics simulations and Laue diffraction, complemented by analytical crystallographic theory analysis. The authors find a classically 'forbidden' transformation mechanism involving a highly strained interface facilitated by internal defects, a transformation that has been previously thought to be not possible. The work posits new dynamical phenomena in solid state phase transitions with potential for strong impact into the community's view of solid state behaviour.

- The authors stress in the introduction that MD simulations that have previously explored the Si-I to Si-II phase transition did not reveal any twinning. In the present paper, the authors have, however, observed twinning in their MD simulations. For a general journal as Nat Commun, it would be helpful if the authors could emphasize why their models differ so significantly from previous works.

- On Pg 3 I believe the authors have made a typo in the cubic coordinates for the Si-II transformational deformation gradients.

- The authors demonstrate how non-traditional, strained interfaces can exist as a result of the dissipation of excess strain energy into the bulk. This poses an interesting question: how far into the bulk does the stress need to dissipate to allow this mechanism to occur, and hence how does the distribution of dislocations influence on the proposed mechanism? Similar topics have been discussed in terms of topotactical transformations and feedback mechanisms in solid state reactivity. On the same idea, could the authors outline their rationale for the use of two dislocations (distance and orientation?) in the simulation of Si-I, do they behave the same? What would be the effect of placing these dislocations closer together (I note the authors state to have tried to maximise the distance between the dislocations)?

- The authors state that the onset pressure for the phase transition is 1.45 times lower for dislocation containing Si-I (ca 6-7 GPa) than for pure Si-I (12 GPa). It does not seem clear from the text at what pressures the authors have observed the experimental transition, and hence how closely this depression in onset pressure correlates (although the authors mention that there is a qualitative correspondence). It seems the DAC was mounted at 12.3 GPa, and the pressure 'increased', although by how much is not clear. I note that this mounting pressure is already twice that for the simulated phase transition temperature – can the authors comment on this significant discrepancy?

- I am a bit surprised to see that the authors have elected to study a martensitic phase transition at 1 K. Certainly, atomic vibrations play a critical role in the mechanism and structure of a transition. Although it is remarkable that there exists still significant correlation of the simulation and experiment, the authors have eliminated a significant degree of freedom from the material in their choice of methodology. Can the authors provide any further justification for their choice, and at least some discussion of the effects of thermal motion on the observed transitions. The authors do state in the SI that there are discrepancies between the MD and experiment, but do not discuss the role of thermal effects in any way. For example, the authors acknowledge in the SI the importance of thermal fluctuations, but have not considered their influence on their atomic model. In my view this is a significant aspect which needs to be addressed more clearly.

I also have a few requests for the authors to address regarding the description of the experimental procedures:

- How quickly was the pressure increased, and was there any evidence for a rate-dependent influence on the propagation of the transformation?
- PTMs of gases have been shown to penetrate into structures during compression (e.g. <https://doi.org/10.1073/pnas.1102361108>). Is there any evidence of such influence within the defects in the present study?

Reviewer #2:

Remarks to the Author:

This work reported an in-situ real-time Laue x-ray diffraction study on the pressure-induced phase transformation in Si. The authors observed $\{111\}$ and $\{110\}$ interfaces in Si-I, which “do not appear in traditional crystallographic theory”. Although the results are quite interesting, one major concern is about the nature of the Si-I to II phase transformation. The crystallographic theory used in this paper is based on the assumption that the phase transformation is a martensitic transition. However, the Si-I to II phase transformation investigated in this work is usually considered as a reconstructive phase transition instead of a martensitic transition in many previous studies, e.g., Rev. Mod. Phys. 75, 863 (2003), APL 113, 123103 (2018), Nature Phys 15, 89–94 (2019), etc. This fact obviously affects the major conclusions of the manuscript. Therefore, I do not recommend the manuscript for publication in Nature Communications.

The authors' response to the Reviewers' comments

REVIEWER COMMENTS

Reviewer #1 (Remarks to the Author):

Reviewer's comment:

Thank you for the opportunity to review the manuscript by Chen and colleagues. The paper describes a detailed investigation of the Si-I to Si-II pressure-induced phase transformation through a combination of molecular dynamics simulations and Laue diffraction, complemented by analytical crystallographic theory analysis. The authors find a classically 'forbidden' transformation mechanism involving a highly strained interface facilitated by internal defects, a transformation that has been previously thought to be not possible. The work posits new dynamical phenomena in solid state phase transitions with potential for strong impact into the community's view of solid state behaviour.

Authors' Response:

We greatly appreciate the Reviewer's positive evaluation of our results.

Reviewer's comment:

- The authors stress in the introduction that MD simulations that have previously explored the Si-I to Si-II phase transition did not reveal any twinning. In the present paper, the authors have, however, observed twinning in their MD simulations. For a general journal as Nat Commun, it would be helpful if the authors could emphasis why their models differ so significantly from previous works.

Authors' Response:

We slightly modified or sentence in p. 6 to address the Reviewer's concern more directly:

“In previous MD studies¹⁰⁻¹² and our studies without dislocations, single variant Si-II was reported only, i.e., stress concentrator at dislocations is responsible for twinning and the completely different microstructure.”

Reviewer's comment:

- On Pg 3 I believe the authors have made a typo in the cubic coordinates for the Si-II transformational deformation gradients.

Authors' Response:

The Reviewer is right, and we corrected a typo. Now the transformation strain looks like in the Supplementary Material.

Reviewer comment:

- The authors demonstrate how non-traditional, strained interfaces can exist as a result of the dissipation of excess strain energy into the bulk. This poses an interesting question: how far into the bulk does the stress need to dissipate to allow this mechanism to occur, and hence how does the distribution of dislocations influence on the proposed mechanism? Similar topics have been discussed in terms of topotactical transformations and feedback mechanisms in solid state reactivity.

On the same idea, could the authors outline their rationale for the use of two dislocations (distance and orientation?) in the simulation of Si-I, do they behave the same? What would be the effect of placing these dislocations closer together (I note the authors state to have tried to maximise the distance between the dislocations)?

Authors' Response:

We added the following text in the Method section:

“We are interested in introducing a single dislocation as a nucleation site for initiation of the Si-I – Si-II PT. However, to minimize its effect on the periodic boundary conditions at the boundaries, dislocation dipole must be placed with equal in magnitude but opposite Burgers vectors²³⁻²⁴. To reduce the effect of one dislocation on the stress concentrator from another one, distance between dislocations should be maximized.”

It was also written few lines below that

“The dislocation dipole was put on the diagonal of the yz plane to maximize the distance of the two dislocations. One of the simulation cells has dimensions of $L_y \approx 99.7$ nm and $L_z \approx 130$ nm and the distance between dislocations of about 82.01 nm. The length L_x has been varied

at 1 K from 4 nm to 30 nm, with a sample containing around 2.5 million to 15 million atoms, and the results are found to be independent of this length due to the periodic boundary conditions applied. Another simulation cell has dimensions of $L_y \approx 199.4$ nm, $L_z \approx 260$ nm, and $L_x \approx 4$ nm, with the distance between dislocations of 164.02 nm containing around 10 million atoms and the microstructure and the obtained at 300 K nanostructure and pressure for initiation of the PT were the same as for smaller sample and 1 K. Thus, the effect of one dislocation on the nucleation processes at another dislocation is negligible, and our results can be interpreted as nucleation and growth at a single dislocation.”

Concerning the question “how far into the bulk does the stress need to dissipate to allow this mechanism to occur?” We can comment the following. Finite element simulations show that high elastic stresses and energy from the twinned interface practically do not penetrate into the bulk but localize near and between the alternating twin tips. This energy, along with surface energy, determines twin spacing and tips rearrangement, like splitting and blunting. It is difficult to evaluate them analytically, especially for the Si-I - twinned Si-II interface, because of complex geometry, large strains, and strong nonlinearity. Also, the main contribution to the bulk energy comes not from interfaces but from the large volume change in the entire Si II nucleus, which strongly propagates into bulk, roughly like $1/r$, where r is the distance from the center of the nucleus. Also, the energy of the new nanostructure should be evaluated, which is practically impossible to do analytically. Thus, an analytical criterion is currently impossible. MD also cannot help because the main parameters (e.g., twin spacing and tip structure) cannot be varied independently. This is indeed a very important but complex problem, which we may consider in nearest future.

Reviewer’s comment:

The authors state that the onset pressure for the phase transition is 1.45 times lower for dislocation containing Si-I (ca 6-7 GPa) than for pure Si-I (12 GPa). It does not seem clear from the text at what pressures the authors have observed the experimental transition, and hence how closely this depression in onset pressure correlates (although the authors mention that there is a

qualitative correspondence). It seems the DAC was mounted at 12.3 GPa, and the pressure ‘increased’, although by how much is not clear. I note that this mounting pressure is already twice that for the simulated phase transition temperature – can the authors comment on this significant discrepancy?

Authors’ Response:

The Reviewer slightly misunderstood our statement

“In MD simulations with two dislocations pressure for initiation of Si-I→Si-II PT is lower than for dislocation-free crystal by a factor of 1.45. Strong effect of defects on nucleation pressure qualitatively corresponds to our experiments, because there are parts of a sample where nucleation did not occur (supplementary information).”

We do not compare the PT pressure from MD and experiment, just from MD simulations with and without dislocations. Pressure from MD under hydrostatic loading is much larger than in experiment, in particular, because to run PT during 40 ps in MD, which takes 38 minutes in the experiments, one needs a much larger thermodynamic driving force and, consequently, pressure. The crystallographic theory does not include pressure at all. We mentioned in the introduction and the concluding remarks the differences in conditions between MD, XRD, and crystallographic theory, stressing that despite all these differences, at least the new nanostructure is consistent with all three methods. Motivated by the Reviewer’s comment, we showed that relative reduction in pressure in MD and experiment is the same and added the following text:

“In MD simulations with a dislocation dipole (practically two independent single dislocations), pressure for initiation of Si-I→Si-II PT is lower than for dislocation-free crystal by a factor of 1.45. In reality, the defect-free Si-I should transform to Si-II at 18.3 GPa, when the first phonon instability in the first principle calculations is observed⁹. Then the observed PT pressure in the current experiment, ~13 GPa (see supplementary material), which matches the typically reported pressures^{3,5-9}, is lower than 18.3 GPa by a factor of 1.41. Thus, a typical nucleation event in the experiment should occur at a single dislocation, consistent with low dislocation density in our Si-I sample. This contrasts with the statement in⁹ that due to the large difference between calculated PT pressure 18.3 GPa and 13 GPa in the experiment, phonon instability cannot be responsible for initiation of PT; clearly, the effect of dislocations was missed. Another

qualitative confirmation of the strong effect of defects on nucleation pressure in our experiments comes from the fact that there are irregular parts of a sample where nucleation did not occur at all because they are probably dislocation-free (see supplementary information).”

We also added the following text to paragraph 5.1 of supplementary information:

“Since pressure for Si-I to Si-II PT is well known and PT occurs at about 13 GPa (see references [3,5-9] in the main text), we did not focus on precise detection of PT pressure in this study and did not use online Ruby fluorescence system to monitor pressure.”

and

“After about 14 hours, the first changes in the sample due to the PT transition were observed, and membrane pressure was not increased any more. The transition was mainly completed in 38 minutes, although a small piece of the initial Si-I sample existed even more than 4 hours after the transition in the other parts of the studied area was finished (see below). The pressure was measured again with the Ruby system, 22 hours after the data collection procedure started, and found to be 13.8 GPa. Therefore, the pressure rate was less than 0.1 GPa/hour in average; although it was higher before the transition (as membrane pressure was increased) and, most likely, pressure still slightly increased also after the transition (as typically DACs have some pressure drifts even without an increase of membrane pressure). Note that the current pressure controlling system cannot control pressure with better precision than a few kbar/hour. Thus, PT occurred at ~13 GPa, and the estimated pressure increase during 38 minutes between initiation and completion of PT should not exceed 0.1 GPa. The pressure range and pressure rate here are like that in [13] (0.2-0.3 GPa/hour); thus, combining results from both experiments is legitimate.”

Reviewer’s comment:

- I am a bit surprised to see that the authors have elected to study a martensitic phase transition at 1 K. Certainly, atomic vibrations play a critical role in the mechanism and structure of a transition. Although it is remarkable that there exists still significant correlation of the simulation

and experiment, the authors have eliminated a significant degree of freedom from the material in their choice of methodology. Can the authors provide any further justification for their choice, and at least some discussion of the effects of thermal motion on the observed transitions. The authors do state in the SI that there are discrepancies between the MD and experiment, but do not discuss the role of thermal effects in any way. For example, the authors acknowledge in the SI the importance of thermal fluctuations, but have not considered their influence on their atomic model. In my view this is a significant aspect which needs to be addressed more clearly.

Authors' Response:

We performed MD simulations at 300 K and obtained the same results. We added in the Methods section:

“Simulations have been carried out at two constant temperatures, 1 K and 300 K, using the Nosé-Hoover thermostat, and for multiple geometries; the obtained nanostructure and even pressure for initiation of PT were the same.”

We also performed simulations for 4 times larger sample and two times larger distance between dislocations at 300 K and again obtained the same results.

Reviewer's comment:

I also have a few requests for the authors to address regarding the description of the experimental procedures: How quickly was the pressure increased, and was there any evidence for a rate-dependent influence on the propagation of the transformation?

Authors' Response:

We addressed this question in response to the previous remark and Section 5.1 of supplementary material. We combined results from only two experiments, which intentionally have been conducted with similar small pressure rates, which are within error of pressure controlling system. Therefore, we are not considering any experimental evidence for pressure rate dependence of the transition.

Reviewer’s comment:

PTMs of gases have been shown to penetrate into structures during compression (e.g. <https://doi.org/10.1073/pnas.1102361108>). Is there any evidence of such influence within the defects in the present study?

Authors’ Response:

We added the following text in Section 5.1 of the supplementary material:

“While it is mentioned in [5] that He penetrates into SiO₂ glass and essentially changes its compressibility and is detected in Raman spectra, we did not find any mentioning of such effects for Si-I in the huge existing literature. Even in [5], neither GeO₂ glass nor crystalline phases of SiO₂ demonstrated this phenomenon.”

Reviewer #2 (Remarks to the Author):

Reviewer’s comment:

This work reported an in-situ real-time Laue x-ray diffraction study on the pressure-induced phase transformation in Si. The authors observed {111} and {110} interfaces in Si-I, which “do not appear in traditional crystallographic theory”. Although the results are quite interesting, one major concern is about the nature of the Si-I to II phase transformation.

Authors’ Response:

We glad that the Reviewer find our results interesting.

Reviewer’s comment:

The crystallographic theory used in this paper is based on the assumption that the phase transformation is a martensitic transition. However, the Si-I to II phase transformation investigated in this work is usually considered as a reconstructive phase transition instead of a martensitic transition in many previous studies, e.g., Rev. Mod. Phys. 75, 863 (2003), APL 113, 123103 (2018), Nature Phys 15, 89–94 (2019), etc. This fact obviously affects the major conclusions of the manuscript. Therefore, I do not recommend the manuscript for publication in Nature Communications.

Authors' Response:

This is a pure terminological misunderstanding, which does not affect any of our result and conclusions.

Since this is a quite common problem and to address the Reviewer's concern, we added in the Supplementary Materials the following new Section:

“6. Reconstructive versus martensitic phase transformation from Si-I to Si-II

The potential problem often arises from the different classifications of structural PTs in different communities. One of the communities, which focuses on bonding between atoms, calls PT a reconstructive if it involves breaking/changing the bonds; see the book [16]. Alternative PTs without bond breaking are often called displacive. Then the PT from the semiconductive Si-I to metallic Si-II clearly belongs to reconstructive PTs [16-19]. There are some other features of these groups of PTs, e.g., reconstructive PT involves large displacements, and no group-subgroup relationship exists between phases, while for displacive PTs, displacements are small, and phases obey group-subgroup relationship [16]. Communities that are concerned with material science, crystallographic, and microstructural aspects do not care about bonding and divide structural PTs into diffusive and diffusionless or martensitic [2,20-23]. Diffusive PTs are accompanied by diffusion and exchange of atomic neighbors during PT, like amorphization and PT occurring via intermediate disordered/amorphous state, precipitation, eutectoid, and massive PT, as well as ordering. Martensitic PTs, in contrast, do not involve diffusion and exchange of atomic neighbors, and mapping of positions of two lattices can be presented by homogeneous deformation (transformation deformation gradient) and some additional intra-cell displacements called shuffles or shifts. Twinning is considered a particular case of martensitic PT, for which the same lattices are connected by a transformation deformation gradient representing simple shear along the twinning plane. Internal stresses and evolution of martensitic microstructure are determined by minimization of the elastic energy, which, in addition to an external load, is completely determined by the field of the transformation deformation gradient and is independent of shuffles. That is why the main parameter in the crystallographic theory and theory of the microstructure is the transformation deformation gradient [2,23], independent of breaking or not breaking bonds. These theories determine, based on the transformation deformation gradient and assumed mode of the lattice-invariant shear (slip or twinning), normal

to the habit and twinning planes, the volume fraction of twin-related martensitic variants (see Eqs. (S1)-(S2)), and orientation relationship between lattices of different phases [2,23].

Most of the reconstructive PTs discussed in the monograph [16] are described by the crystallographic theory of martensitic PTs, including PT in iron [23], shape memory alloys [2], plutonium [24], and hexagonal and rhombohedral graphite to hexagonal and cubic diamond and similar PTs in BN [25,26]. It is mentioned in [26] that graphite-like phases of BN have very strong covalent bonds with sp^2 hybridization within the hexagonal planes and weak van der Waals bonds between these planes, while cubic or wurtzitic superhard BN have only covalent (partially ionic) bonds, with tetrahedral three-dimensional sp^3 hybridization; the same is true for graphite and diamond. Despite the reconstructive PT between these phases, all aspects of crystallographic and microstructure formation theory are applied in [25,26] to these PTs; they are called martensitic in [25].

With a full understanding of change in bonding during PT Si-I – Si-II in [1,27], it is treated as martensitic PT in which the transformation deformation gradient connects atoms of Si-I and Si-II lattice even without shuffles. All continuum theories of PTs involving the transformation deformation gradient (e.g., based on energy minimization [2] or phase field theories [28-31], including those for Si [28-30]) do not distinguish bonding. However, some works distinguish between reconstructive martensitic PTs, for which no group-subgroup relationship exists between phases, and “weak” martensitic PTs, for which phases obey group-subgroup relationship [31,32]. Reconstructive martensitic PTs are more complex for modeling because the reverse PT may occur to different variants of the austenite related by a lattice-invariant shear. But this is not the case for cubic to tetragonal PT Si-I – Si-II.

To summarize, various definitions of martensitic and reconstructive PTs are used by different research communities depending on their goals. For example, the firefighters distinguish a violin from a piano in a simple way: the piano burns longer, which is sufficient for their goals. Since our main goal here is crystallography and microstructure rather than atomic bonding, we call Si-I to Si-II PT martensitic. Our results and conclusions would not change if we called them reconstructive or by any other name.”

Reviewers' Comments:

Reviewer #1:

Remarks to the Author:

Thank you for the opportunity to review the revised manuscript by Chen and colleagues. The authors have made extensive efforts to address the reviewer comments and have significantly revised the text accordingly. In my opinion the text is now much more accessible to the wider audience of Nat Commun and provides a rigorous discussion of these interesting findings. This is a very interesting paper that I am certain will provoke interesting discussion and science in the community.

It is clear from the text and the author responses that there are indeed many interesting questions left to be answered regarding the Si phase transitions, ensuring continuity of this research direction.

Reviewer #2:

Remarks to the Author:

The response and revision have addressed my major concern.

There is a relevant paper on the mechanism of the Si-I to II phase transformation which was missed by the authors [Hannelore Katzke et al., PHYSICAL REVIEW B 73, 134105 (2006)]. It would be helpful to discuss how the current work agrees or disagrees with the previous results.

The authors' response to the Reviewers' comments

REVIEWER COMMENTS

Reviewer #1 (Remarks to the Author):

Reviewer's comment:

Thank you for the opportunity to review the revised manuscript by Chen and colleagues. The authors have made extensive efforts to address the reviewer comments and have significantly revised the text accordingly. In my opinion the text is now much more accessible to the wider audience of Nat Commun and provides a rigorous discussion of these interesting findings. This is a very interesting paper that I am certain will provoke interesting discussion and science in the community.

It is clear from the text and the author responses that there are indeed many interesting questions left to be answered regarding the Si phase transitions, ensuring continuity of this research direction.

Authors' Response:

We greatly appreciate the Reviewer's positive evaluation of our results. We will continue working in this direction.

Reviewer #2 (Remarks to the Author):

Reviewer's comment:

The response and revision have addressed my major concern.

Authors' Response:

We greatly appreciate the Reviewer's positive evaluation of our results.

Reviewer's comment:

There is a relevant paper on the mechanism of the Si-I to II phase transformation which was missed by the authors [Hannelore Katzke et al., PHYSICAL REVIEW B 73, 134105 (2006)]. It would be helpful to discuss how the current work agrees or disagrees with the previous results.

Authors' Response:

We cited this interesting paper along with other previously cited papers, which give similar relevant information (the transition pressure (12-13 GPa), volume drops (23.7%), etc. However, the authors do not consider martensitic nanostructure, so it cannot be compared.